# Consequences of the Lack of TNFR1 in Ouabain Response in the Hippocampus of C57BL/6J Mice

**DOI:** 10.3390/biomedicines10112937

**Published:** 2022-11-15

**Authors:** Paula Fernanda Kinoshita, Ana Maria Orellana, Diana Zukas Andreotti, Giovanna Araujo de Souza, Natalia Prudente de Mello, Larissa de Sá Lima, Elisa Mitiko Kawamoto, Cristoforo Scavone

**Affiliations:** 1Laboratory of Molecular Neuropharmacology, Department of Pharmacology, Institute of Biomedical Sciences, University of São Paulo, São Paulo 05508-900, Brazil; 2Laboratory of Molecular and Functional Neurobiology, Department of Pharmacology, Institute of Biomedical Sciences, University of São Paulo, São Paulo 05508-900, Brazil

**Keywords:** TNF, TNFR1, TNFR2, ouabain, behavior, neuroinflammation, glutamate receptors

## Abstract

Ouabain is a cardiac glycoside that has a protective effect against neuroinflammation at low doses through Na^+^/K^+^-ATPase signaling and that can activate tumor necrosis factor (TNF) in the brain. TNF plays an essential role in neuroinflammation and regulates glutamate receptors by acting on two different receptors (tumor necrosis factor receptor 1 [TNFR1] and TNFR2) that have distinct functions and expression. The activation of constitutively and ubiquitously expressed TNFR1 leads to the expression of pro-inflammatory cytokines. Thus, this study aimed to elucidate the effects of ouabain in a TNFR1 knockout (KO) mouse model. Interestingly, the hippocampus of TNFR1 KO mice showed a basal increase in both TNFR2 membrane expression and brain-derived neurotrophic factor (BDNF) release, suggesting a compensatory mechanism. Moreover, ouabain activated TNF-α-converting enzyme/a disintegrin and metalloprotease 17 (TACE/ADAM17), decreased N-methyl-D-aspartate (NMDA) receptor subunit 2A (NR2A) expression, and induced anxiety-like behavior in both genotype animals, independent of the presence of TNFR1. However, ouabain induced an increase in interleukin (IL)-1β in the hippocampus, a decrease in IL-6 in serum, and an increase in NMDA receptor subunit 1 (NR1) only in wild-type (WT) mice, indicating that TNFR1 or TNFR2 expression may be important for some effects of ouabain. Collectively, our results indicate a connection between ouabain signaling and TNFR1, with the effect of ouabain partially dependent on TNFR1.

## 1. Introduction

Ouabain is a cardiotonic steroid hormone produced in the adrenal and pituitary gland [1,2,3,4,5] that interacts with Na^+^/K^+^-ATPase (NKA), a highly conserved membrane protein that plays an important role in cell osmotic balance and is critical for neuronal excitability [6,7]. NKA has two distinct pools. One is related to ion transport, whereas the other is related to protein–protein interactions and is present primarily in lipid rafts, leading to the activation of signaling pathways independent of Na^+^ and K^+^ levels [8]. Interestingly, ouabain acts in these two distinct pools depending on its concentration. However, little is known about the physiological functions of ouabain in the organism, particularly in the brain.

High doses of ouabain can result in cellular death through the inhibition of NKA [9], whereas low doses of ouabain are protective [10]. It has been demonstrated that the activation of brain-derived neurotrophic factor (BDNF) and nuclear factor-kappa B (NF-κB) by low doses of ouabain provokes changes in the hippocampal environment, promoting dendritic arborization [11]. These changes improve spatial memory and inhibit long-term memory extinction, exhibiting the fundamental role of NKA and ouabain in central nervous system (CNS) functional changes [12]. Moreover, ouabain has been demonstrated to have anti-inflammatory effects on the CNS [13]. Previous studies by our group showed that ouabain reduced pro-inflammatory cytokines in the hippocampus when rats and primary glial cell cultures were exposed to lipopolysaccharides (LPS) [14] and that the α2 NKA isoform (α2-NKA) appears to be essential for the LPS-induced inflammatory response [15], suggesting that ouabain and NKA interaction can modulate genes involved in the innate immune response.

Ouabain has also been shown to increase tumor necrosis factor (TNF) levels in the hippocampus and cerebellar cells of rats by activating NF-κB through the N-methyl-D-aspartate (NMDA) signaling pathway [16,17]. Similar results were observed in retinal cells, where ouabain increased their survival rate by augmenting TNF and interleukin (IL)- 1β levels [18,19]. TNF is a pleiotropic cytokine [20] essential for the innate immune response against pathogens. TNF is synthesized as a transmembrane protein (tmTNF) that is converted into soluble TNF (solTNF) by TNF-α-converting enzyme/a disintegrin and metalloprotease 17 (TACE/ADAM17) cleavage. These two forms of TNF are biologically active and produced by microglia, astrocytes, and some types of neurons in the CNS. The production of each TNF is dependent on the cell type and type of stimulus, TACE/ADAM17 activity, and the presence/activity of TACE inhibitors [21,22].

Chronic exposure to TNF can be deleterious, causing microglial overactivation and damage to neuronal function, as well as neurogenesis and necroptosis [23,24,25]. In neurodegenerative diseases, such as Alzheimer’s disease (AD) [26,27], multiple sclerosis [28], and Parkinson’s disease (PD), the levels of TNF are increased in the post-mortem brain, serum, and cerebrospinal fluid (CSF) [29,30]. Therefore, TNF has become an important target for treating neurodegenerative diseases.

The use of TNF blockers in patients with rheumatoid arthritis (RA) and psoriasis is associated with a lower risk of AD [31]. In addition, patients with AD treated with etanercept and infliximab showed cognitive improvement [32,33,34]. However, there are still issues related to blood–brain barrier (BBB) penetration for these agents and the stage at which the treatment is performed [33]. The use of TNF blockers is not applied in multiple sclerosis treatment because of its association with demyelination [35,36,37].

One of the reasons for the failure of anti-TNF treatment is likely due to variations in TNF receptors. TNF binds to two receptors: tumor necrosis factor receptor 1 (TNFR1) and TNFR2. TNFR1 is expressed in all cells and can be activated by both forms of TNF, with a preference for solTNF. TNFR2 is mainly expressed in immune system cells, microglia, endothelial cells, and some neuronal populations, and is only activated by tmTNF [29]. TNFR1 and TNFR2 may have antagonistic or synergistic actions depending on the context [38].

TNFR1 has a death domain that binds to TNFR1-associated death domain protein (TRADD), leading to NF-κB activation, which initiates pro-survival signaling, proliferation, pro-inflammatory cytokines, and apoptosis pathways. TNFR2 does not contain a death domain and, as a result, activates different pathways related to inflammation and survival [29]. TNFR2 can also activate the phosphoinositide 3-kinase/protein kinase B (PI3K/Akt) pathway, leading to cell proliferation and survival [39,40]. Interestingly, TACE/ADAM17 also cleaves TNFRs and releases soluble TNFR (sTNFR), which sequesters circulating TNF and decreases TNFR availability in the membrane [41,42,43].

It has been shown that TNF is important for the modulation of synapses in the CNS. TNF is important for synaptic plasticity in a phenomenon called homeostatic synaptic plasticity, which allows the homeostasis of neuronal activity in a range optimal for neurotransmission [44,45]. This phenomenon is a response to high chronic levels of neuronal activity through negative feedback, which decreases the firing rate of action potentials [46]. This control appears to be mostly modulated by astrocytes, altering α-amino-3-hydroxy-5-methyl-4-isoxazolepropionic acid (AMPA) expression and causing an increase in mEPSC amplitude [47].

TNF also acts indirectly in neurons by inhibiting glutamate transport in astrocytes and/or by increasing AMPA and NMDA receptor expression. These modulations increase excitatory synapses potentiated by a decrease in the expression of γ-aminobutyric acid A (GABA_A_) [48].

The TNF and ouabain signaling pathways are complex and not well understood in the CNS. Based on prior work, we hypothesized that ouabain may affect neuroinflammation through TNFR1 signaling. Thus, this study aimed to test this hypothesis by analyzing changes in behavior at the molecular level in TNFR1 knockout (KO) mice treated with ouabain compared to untreated mice.

## 2. Materials and Methods

### 2.1. Animals and Treatment

The experiments with animals were performed according to the guidelines of the Brazilian Society of Laboratory Animals Science (SBCAL) and ARRIVE guidelines and under the norms of the Ethical Committee for Animal Research of the Institute of Biomedical Sciences (CEUA/ICB/USP, protocol no. 37/2014).

C57BL/6J wild-type (WT) or homozygous KO *Tnfr1*^tm1Mak^ (TNFR1 KO) mice were obtained from the Animal Facility of the School of Medicine, University of São Paulo, and housed in the mice room of the Pharmacology Department (ICB-USP) until they were 2–4 months old. They were kept under a 12 h light/dark cycle (lights on at 7:00 a.m.) and allowed free access to food and water. To analyze the influence of ouabain on TNFR1 signaling, both WT and TNFR1 KO mice were randomly assigned to four groups: (1) WT saline, (2) WT ouabain, (3) TNFR1 KO saline, and (4) TNFR1 KO ouabain. They were injected once with ouabain (30 μg/kg, intraperitoneal [i.p.]) [49] or saline (vehicle).

For biochemical experiments, animals were euthanized 2 h after treatment administration, as previously determined by our group [14]. The brains were immediately removed and immersed in cold phosphate-buffer saline (PBS). Each hippocampus was rapidly dissected, immersed in dry ice, and stored at −80 °C until further analysis. All procedures were performed in the morning.

For the behavioral assays, the animals were evaluated in the open field test and the elevated plus maze 24 and 48 h after treatment injection, respectively. A previous study showed that behavioral changes caused by ouabain can still be observed after 7 or 14 d [12]. Only male mice were used in the experiments because of the influence of progesterone as a precursor of ouabain synthesis [50] and the putative interactive action between them [51].

### 2.2. Chemicals and Enzyme-Linked Immunosorbent Assay (ELISA) Kits

Routine reagents and ouabain were purchased from Sigma–Aldrich. The TNF-α (DY410-05), IL-6 (DY406-05), and IL-1β (DY401-05) immunoassay kits were purchased from R&D Systems. A mouse TACE/ADAM17 kit was used to measure ADAM17 activity (BT Lab). BDNF levels were measured using the BDNF Emax^®^ immunoassay system (Promega). All ELISA were performed in accordance with the manufacturer’s instructions. Target protein concentrations were measured in the serum and hippocampus.

### 2.3. Protein Extraction-Membrane Enriched, Cytosolic, and Nuclear Fractions

Membrane-enriched, cytosolic, and nuclear fractions were extracted from the hippocampus [14]. The Dounce homogenizer was used to process the hippocampal structures in cold PBS with inhibitors of protease and phosphatase. The homogenate was centrifuged at 4 °C for 30 s at 12,000× *g*. The supernatant was collected and centrifuged at 4 °C for 20 min at 12,000× *g*. Whereas the pellet was resuspended in a buffer containing: 0.32 M sucrose, 20 mM HEPES, 1 mM EDTA, 1 mM DTT, and 1 mM PMSF (pH 7.4), resulting in a membrane-enriched fraction. The pellet obtained from the first centrifugation was resuspended in lysis buffer supplemented with protease and phosphatase inhibitors and incubated on ice for 10 min. After addition of NP-40 (10%), the samples were mixed and centrifuged for 30 s at 12,000× *g*. The supernatant was reserved for Western blotting and ELISA (cytosolic fraction). The pellet was resuspended in extraction buffer containing:1.5 mM MgCl_2_, 20 mM HEPES, pH 7.9, 25% glycerol, 300 mM NaCl, 0.5 mM PMSF, 0.25 mM EDTA, 2 μg/mL leupeptin, 2 μg/mL antipain, 3 mM sodium orthovanadate, 30 mM NaF, and 20 mM sodium pyrophosphate. The resuspension was kept on ice for 20 min and centrifuged at 4 °C for 20 min at 12,000× *g*. The result supernatant (nuclear fraction) was used in the electrophoretic mobility shift assay (EMSA). The protein concentration was determined by the Bradford technique [52].

### 2.4. Western Blotting

Western blotting assay was prepared as previously described [14]. A 10% polyacrylamide gel and Bio-Rad mini-Protean III apparatus (Bio-Rad) were used to separate the membrane-enriched proteins (15 µg) by electrophoresis (SDS-PAGE; 90 V). The proteins were blotted onto a nitrocellulose membrane (Bio-Rad) and incubated overnight with the specific primary antibodies (1:1000) TNFR2 (sc-7862, Santa Cruz Biotechnology), NR2A, pAMPA, AMPA, and NR1 (4205, 8084, 13,185, and 5704, Cell Signaling Technology). The membranes were then incubated for 2 h with a specific secondary antibody.

Proteins recognized by antibodies were detected using electrochemiluminescence (ECL). To standardize and quantify the immunoblots, we used ImageJ software (NIH). The β-ACTIN antibody (sc-1616; Santa Cruz Biotechnology) was used as an internal experimental control, with results expressed concerning β-ACTIN intensity.

### 2.5. EMSA

EMSA was performed as previously described [12,15]. Briefly, double-stranded oligonucleotides containing NF-κB (5′-AGTTGAGGGGACTTTCCCAGGC-3′) or CREB (5′-AGAGATTGCCTGACGTCAGAGAGCTAG-3′) consensus sequences were labeled with radioactive γ-^32^P dATP. Nuclear extracts were incubated with one of the sequences and then incubated with a reaction buffer for 30 min at room temperature. Sample-NF-κB or CREB complexes were size-separated using electrophoresis through a 6% acrylamide:bis-acrylamide (37.5:1) gel in TBE for 2  h at 150 V. Gels were vacuum-dried for 1  h at 80 °C and exposed to X-ray film at −80 °C for a week before revelation. Autoradiographs were visualized using a photo documentation system (DP-001-FDC) and quantified using ImageJ software (NIH).

### 2.6. Open Field

In this test, locomotion, exploration, anxiety-like behavior, and fear behavior can be evaluated. Twenty-four hours after the injection with ouabain or saline, the animals were allowed to explore a plastic cage (35 × 40 × 15 cm) for 10 min, and some parameters were recorded, such as distance traveled, time spent in the center and periphery, and mean speed for the first 5 min. The activity of the mice was analyzed using ANY-maze image software. Between tests, the cages were cleaned with ethanol.

### 2.7. Elevated Plus-Maze

Twenty-four hours after the open field test, the elevated plus-maze test was performed to observe whether the lack of TNFR1 or treatment can change anxiety-like behavior. The test consisted of placing the animal in the central area of a cross-shaped maze with two types of arms (open and closed) for 5 min. The number of entries and the time spent in each type of arm were measured using the ANY-maze software. The time in the center was quantified but not used in the analysis. The scheme of the behavioral test is shown in Appendix A, along with the number of animals used for each experiment. All the animals were genotyped before treatments (Appendix A).

### 2.8. Statistical Analysis

The results were expressed as mean  ±  SEM of the indicated number of experiments. Statistical comparisons for Western blotting, EMSA, ELISA, and behavioral tests were mostly performed using two-way analysis of variance (ANOVA), followed by Tukey’s post-hoc test. All analyses were performed using the Prism 7 software package (GraphPad Software, San Diego, CA, USA). Statistical significance was set at *p* < 0.05.

## 3. Results

### 3.1. TNFR1 KO Mice Have Altered TNFR2 and Ouabain Increases TACE/ADAM17 Activity

First, it was investigated whether ouabain (30 μg/kg) can change TNF levels and/or the TNF signaling pathway in mice. In the serum (Figure 1A) and hippocampus (Figure 1B) samples, there was no difference between the groups. However, basal membrane expression of TNFR2 was increased in TNFR1 KO animals, even in the presence of ouabain. This result suggests a compensation for the lack of TNFR1 expression (Figure 1C). To investigate TNFR1 and TNFR2 expression and TNF cleavage, we measured the activity of TACE/ADAM17, an essential enzyme that regulates these processes. Ouabain increased TACE/ADAM17 activity independently of TNFR1 expression (Figure 1D).

### 3.2. Ouabain Has an Effect on IL-1β Expression That Is Correlated with NF-кB Activity

Ouabain can also activate IL-1β [15,16,19,53,54], a pro-inflammatory cytokine with inflammasome-dependent release [55] which exhibits different pathway activation and timing that may be affected by the lack of TNFR1. IL-1β can regulate TNF responses, especially by increasing the amount of membrane TNFR [56] and increasing TNF levels through NF-kB activity [57]. However, IL-1β and TNF activate NF-kB and regulate miR-146 [58,59]. Both are important in the immune response and are the major cytokines released by microglia in neurodegenerative diseases and trauma.

In the serum, TNFR1 KO mice had increased basal levels of IL-1β compared to WT mice, and ouabain treatment reversed this effect (Figure 2A). Interestingly, in the hippocampus, WT mice in the ouabain-treated group showed an increase in IL-1β compared to those in the control group (Figure 2B), as previously reported [15]. These differences in IL-1β may be related to timing differences, genotype effects, and secondary effects of ouabain in the periphery and in the brain, or the brain may respond differently to ouabain than the periphery.

IL-6 levels in the serum and hippocampus were also evaluated. In the serum, only a decrease in the basal levels of IL-6 was caused by ouabain in WT mice (Figure 2C), whereas there was no difference between the basal levels of IL-6 in WT and TNFR1 KO mice. However, no changes in IL-6 levels were found in the hippocampus between the groups (Figure 2D). The activity of NF-кB was also measured in the hippocampus, as this transcription factor plays a crucial role in inflammation (Figure 2E). The activity of NF-кB decreased in TNFR1 KO mice when they were treated with ouabain, which correlated with the lack of increase in IL-1β in TNFR1 KO mice treated with ouabain (Figure 2B).

### 3.3. TNFR1 Impacts BDNF Levels in Serum and Ouabain Modulates Glutamate Receptors

The activity of CREB (Figure 3A) and level of BDNF (Figure 3B) were measured in the hippocampus, and an interaction between both genes and ouabain treatment was detected. Interestingly, the levels of BDNF in serum (Figure 3C) were increased in TNFR1 KO animals, independent on ouabain treatment.

TNF interferes with the glutamatergic system. TNF can upregulate NR1 phosphorylation [60], and TNFR1 is involved in AMPA receptor modulation [61]. Thus, the phosphorylation of AMPA receptor and the expression of NR1 and NR2A were evaluated. The TNFR1 KO mice treated with ouabain exhibited significantly higher phosphorylation of the AMPA receptor than the other groups (Figure 4A). We also observed the expression of total AMPA, but there was no significant change in its expression (Appendix A). Ouabain also decreased NR2A membrane expression independently of TNFR1 gene expression (Figure 4B) and increased NR1 membrane expression in WT mice compared to the TNFR1 KO group treated with ouabain (Figure 4C).

### 3.4. Ouabain Interferes with Behavior Independently of the Presence of TNFR1

Open field and elevated plus-maze tests were performed to determine whether there were functional changes in the animals after treatment with ouabain. In the open field test, the distance traveled, mean speed, and time spent in the center were decreased through ouabain treatment in WT and TNFR1 KO mice. A general effect of ouabain was observed, independent of gene factor (Figure 5A–C). However, no significant change over time was observed in the periphery (Figure 5D). In the elevated plus maze, no difference in time spent in the open arms (Figure 5E) was observed, but there was an increase in time spent in the closed arms with ouabain treatment, independent of gene factor (Figure 5F).

## 4. Discussion

TNF is an important inflammatory mediator that plays a role in homeostasis by controlling synaptic plasticity [62]. Anti-TNF antibody therapy was developed to decrease inflammation in a variety of diseases such as inflammatory bowel disease (IBD), psoriasis, psoriatic arthritis, RA, ankylosing spondylitis, and juvenile idiopathic arthritis [63], emphasizing the role of TNF signaling in several illnesses [42]. However, the TNF response is more complex and refined because of the presence of its receptors (TNFR1 and TNFR2), which have different responses and affinities. Although TNFR1 has been described as deleterious, this receptor is necessary for the regulation of neuronal plasticity and for the repair and neurogenesis in the hippocampus in a stroke model [64,65].

Our laboratory showed that ouabain (10 nM or 10 μM in 1 μL of saline) administrated directly to rat CA1 of the hippocampus through infusion cannula increases TNF m-RNA levels after 1 h [17]. The same pattern was found in cerebellar neurons treated with ouabain (10 µM) for 2 h [16]. However, an increase in IL-1β in the hippocampus of WT mice was observed in the current study (Figure 2B), there was no change in TNF levels in the serum (Figure 1A) or hippocampus (Figure 1B) in the ouabain-treated group, which may be because of a different time frame, animal model, route of administration, or ouabain concentration. It is important to note that the chosen concentration of ouabain was 30 μg/kg, which is 10 times less than the dose used in a past study that did not observe any significant changes in blood pressure, an indicator of NKA inhibition [49]. This dose did not change NKA activity in acute treatment, which was also observed in our results and showed that the effects of ouabain were not related to NKA inhibition (Appendix A). The time point for biochemical analyses was based on an earlier study by our group in rats that showed that intraperitoneal ouabain had an anti-inflammatory effect against LPS 2 h after injection [14].

Our group and others have shown that ouabain at low doses is protective against LPS [14,15,66], increases cell proliferation and growth, and may be crucial for cell development [67]. Two pathways are known to be activated by ouabain: the Ras-Raf-MAPK pathway [68] and IP3R activation that leads to calcium oscillations [69]. However, the impact of TNFR on ouabain has not yet been described. Interestingly, for the first time, we observed that the levels of TNFR2 in the membrane-enriched fraction increased only in the hippocampus of TNFR1 KO mice (Figure 1C), suggesting a compensatory effect in TNF signaling pathways. TNFR2 expression is induced in neurological diseases and is limited to specific cells in the CNS [70,71,72]. TNFR2 plays a protective role in promoting oligodendrocyte progenitor proliferation, remyelination [73], and glutamate excitotoxicity [74]. The increase of TNFR2 in TNFR1 KO mice shows that the TNF response is important for basal activity, such as activating different signaling pathways through a different receptor. 

In hippocampal cells treated with radiation, the lack of TNFR2 led to a decrease in neurogenesis, which did not occur in TNR1 KO or TNF KO mice. The same study showed that TNFR2 in neural stem/progenitor cells promoted growth and/or cell survival, which was inhibited by TNFR1 [75]. The difference between the receptors also appears in AD. TNFR1 levels are increased and TNFR2 levels are decreased in post-mortem brains of patients with AD [76,77]. The affinity of TNF in AD is higher for TNFR1 than for TNFR2 [78]. Deletion of TNFR2 also aggravates the disease, whereas its overexpression rescues this aggravation, showing that the protective effect of TNFR2 is also prominent in patients with AD [79].

TACE/ADAM17 activity was increased in the hippocampus in the presence of ouabain (Figure 1D), but no changes were detected in TNF levels or TNFR2 expression, which showed that there was no correlation with TACE/ADAM17 activity. However, TACE/ADAM17 cleaves other substrates, such as amyloid precursor protein (APP), IL-6 receptor, and toll-like receptor 2 (TLR2), which are mainly related to inflammation and APP in AD [80,81,82]. This evidence shows that ouabain interferes with different pathways that promote protection. It is interesting to note that the increase in TACE/ADAM17 activity was related to an increase in myelinization in a Charcot–Marie–Tooth (CMT) neuropathy model [83], which was recently discovered to be related to mutations in the α1 isoform of NKA [84].

In the serum, TNFR1 KO mice had higher basal IL-1β levels, whereas ouabain decreased IL-1β levels in TNFR1 KO mice, reaching levels similar to those in WT mice (Figure 2A). In the hippocampus, ouabain increased IL-1β levels in WT mice compared to those in the WT saline group (Figure 2B). An increase in IL-1β levels in WT mice was also observed in glial primary cultures and retina cell cultures [19]. This increase in IL-1β was not observed in TNFR1 KO mice, which may be due to the faster response of WT in comparison to TNFR1 KO and/or IL-1β release being TNFR1 dependent. It has been shown that IL-1β levels modulate the TNF response by increasing TNFR2 and sTNFR2; however, this change requires TNFR1 [56].

The differences between the profile of IL-1β levels in the periphery and hippocampus may be because IL-1β can modulate TNF expression, and the increase in Il-1β may be a response to the lack of TNFR1 signaling in the periphery, whereas the control in the hippocampus may be mediated by other factors. A study that analyzed post-mortem human brains of ischemic stroke and blood samples of ischemic stroke patients showed that the profile of cytokines and TNFR expression differed between brain tissue and blood. TNF and IL-1 were expressed by subsets of cells, but TNFR2 was expressed in brain areas with increased astrocytic reactivity, including normal-appearing tissue. Indeed, an increase in brain cytokines was observed, whereas in plasma TNFR1 and TNFR2, TNF-α and IL-1β levels did not change [59].

Cytokine control is also important for modulation of glutamate receptors. In hippocampal neuronal cultures, IL-1β can increase NMDA receptor function and consequently intracellular calcium levels, which are important for cell survival [85]. Thus, the control of IL-1β in the brain requires further refinement. This increase in intracellular calcium occurs through Src, which can also be activated by ouabain. We did not explore other regions of the brain, and this may also be related to the hippocampus.

IL-6 levels in the serum and hippocampus were also evaluated. In the serum, there was a decrease in the basal levels of IL-6 caused by ouabain in WT mice (Figure 2C), whereas no changes in IL-6 in the hippocampus were found (Figure 2D), indicating that the IL-6 response can occur later in the hippocampus or not at all because IL-6 release is dependent on TNF/TNFR1 (Figure 2D) [86]. Recent studies have shown that ouabain can regulate IL-6 in skeletal muscle in a dose-dependent manner. Low doses of ouabain suppressed STAT3 and, consequently, IL-6 signaling, whereas 100 nM ouabain increased the release of IL-6 [87].

The activity of NF-кB in the hippocampus is decreased in TNFR1 KO mice when treated with ouabain, which correlates with no increase in IL-1β levels. Even though TNFR2 has no death domain, its activity can also stimulate NF-кB in a less efficient manner: through TRAF2 with its associated partner TRAF1 [88], which can inhibit non-canonical NF-кB signaling [89]. Our results suggest that the lack of TNFR1 signaling itself cannot alter NF-кB activity without a stimulus (Figure 2E). However, ouabain treatment in TNFR1KO mice decreased NF-кB activity (Figure 2E). Ouabain and the α2 isoform of NKA have been implicated in the modulation of NF-кB activation. Thus, in our model, ouabain acted as a stimulus that, through TNFR2, decreased basal NF-кB, likely in microglia, where TNFR2 is more highly expressed.

Ouabain also modulates CREB nuclear translocation and BDNF release. BDNF in the CNS is an important growth factor related to survival, neuronal differentiation, neurotransmitter regulation, and neuronal plasticity, and is also involved in other processes, such as memory, learning, and behavioral changes. Chronic BDNF deficiency leads to impaired spatial learning in an age-dependent manner [90]. It has been shown in astrocytes that BDNF can be regulated by TNF through NF-κB [91] and TNFR1 KO and TNF KO mice have increased basal lines of neurogenesis, which is dependent on TNFR2 [75]. However, BDNF can also be produced outside the CNS, mostly by platelets in the blood, showing that different mechanisms are involved in the release of BDNF in the periphery versus the brain [92,93]. Collectively, our data suggest an interaction between the gene and treatment in CREB activity (Figure 3A) and BDNF (Figure 3B), with levels that were found to decrease and increase, respectively, in the hippocampus of TNFR1 KO vs. WT mice treated with ouabain. Interestingly, BDNF levels in serum (Figure 3C) were increased in TNFR1 KO mice, which may have an important impact on other tissues and may be related to TNFR2 signal triggering in immune cells. 

Ouabain decreased the expression of the NMDA subunit NR2 membrane independently of TNFR1 expression (Figure 4B) and increased NR1 expression in WT mice compared to TNFR1 KO mice (Figure 4C). Intracerebroventricular (ICV) administration of 0.1 nmol ouabain also increased NR1 expression in the rat hippocampus. In contrast, 10 nmol of ouabain in the same study showed that NR2A expression also decreased in the same administration model [94]. The proximity between NKA isoforms (α1 and α3) and NR2A has been demonstrated in neurons, which may explain why treatment with ouabain decreased NR2A membrane expression. Thus, NKA–NMDA interaction may be responsible for the decrease in NMDA-mediated calcium flux [16,17,95].

The AMPA receptor can be regulated by NKA activity, as NKA is enriched in synapses and associated with AMPA receptors. Inhibition of NKA by high doses of ouabain decreased AMPA receptor expression [96]; however, in our study, low concentrations of ouabain for acute treatment revealed no changes in expression in WT mice. A relationship exists between TNF and homeostatic synaptic plasticity through the modulation of AMPA receptor expression [47]. Another study showed that AMPA receptor surface expression is increased by TNF treatment mediated by neuronal TNFR1 activation [97]. Interestingly, in the present study, ouabain was able to increase the ratio of pAMPA/tAMPA in TNFR1 KO mice (Figure 4A). This result indicates that TNFR1 KO mice are more susceptible to changes in AMPA receptors in the presence of a stimulus. However, there was no difference in AMPA receptor modulation due to TNFR1 absence (Figure 4A), and a study observed that AMPA stimulation of TNFR1 in organotypic culture slices did not result in cell death, as observed in the WT group [98].

The half-life of ouabain when administered intravenously was reported in dogs and humans to be 18 and 21 h, respectively [99]. Another study showed that the half-life of elimination was 23 h in humans [100]. However, its effects on the CNS last longer than 24 h in rodents. A study by our group showed that a single intrahippocampal injection of 10 nm ouabain trigged signaling pathways for 24 h. It increased neuronal NF-kB activation in 1 and 10 h, as well as CREB/BDNF. However, neuronal morphological changes were only significant after 7 and 14 d when we observed an improvement in spatial long-term memory formation [12].

Ouabain plays an important role in the modulation of motor neurons by modulating NKA activity [101]. After 24 h of ouabain treatment, there was a decrease in the total distance traveled in the 5 min of the open field test, which indicates a reduction in environment exploration, corroborating less speed and time in the center. Most studies have shown hyperlocomotion with a high dose of ouabain through ICV injection, which directly inhibits NKA in the brain, leading to a model of mania. Typically, the open field test in these models is performed after a few days of ouabain injection [102,103,104]. However, our open field test without habituation was performed 24 h after injection. A recent study showed a decrease in locomotion within 24 h of ICV injection of ouabain (at the lowest dose of 10 μM tested), supporting our results [105]. The decrease in locomotion also corroborated the reduction in the average speed, which may indicate a reduction in exploration. The animals also spent more time in the peripheral area in the open field test and more time in the closed arms in the elevated plus-maze test, showing a likely increase in anxiety-like behavior caused by ouabain that was maintained even 2 d after an i.p. injection.

The i.p. injection can be considered a limitation of this study. However, from the perspective of the use of ouabain at low concentrations as a neuroprotective drug, i.p. injection is also important to understand how the full organism would respond to this treatment and observe any possible collateral effects or changes in behavior. Another point to discuss is that different regions in the brain can respond differently to ouabain because different percentages of cell types are present in each region, resulting in different levels of alpha subunit expression. ICV injection may also play an important role in the response of the hippocampus, and different regions of the brain interact with the hippocampus. Isolating the response to ouabain injected directly into the hippocampus may be interesting; however, with i.p. injection, we are able to better study the interaction of the full system in the brain.

## 5. Conclusions

Ouabain did not activate TNF in the hippocampus in this model, and some changes occurred only in WT mice, such as the increase in IL-1β and NR1 expression. However, in TNFR1 KO mice, these effects were not TNFR1-dependent and/or blocked by TNFR2 signaling, because TNFR2 and TNFR1 share some signaling pathways. We also revealed that some effects of ouabain are not dependent on TNFR1, such as the increase in TACE/ADAM17 activity and changes in behavior. Thus, the effect of ouabain may be partially mediated by TNFR1. This relationship shows that there is a complexity in ouabain responses that remains to be further elucidated.

## Figures and Tables

**Figure 1 biomedicines-10-02937-f001:**
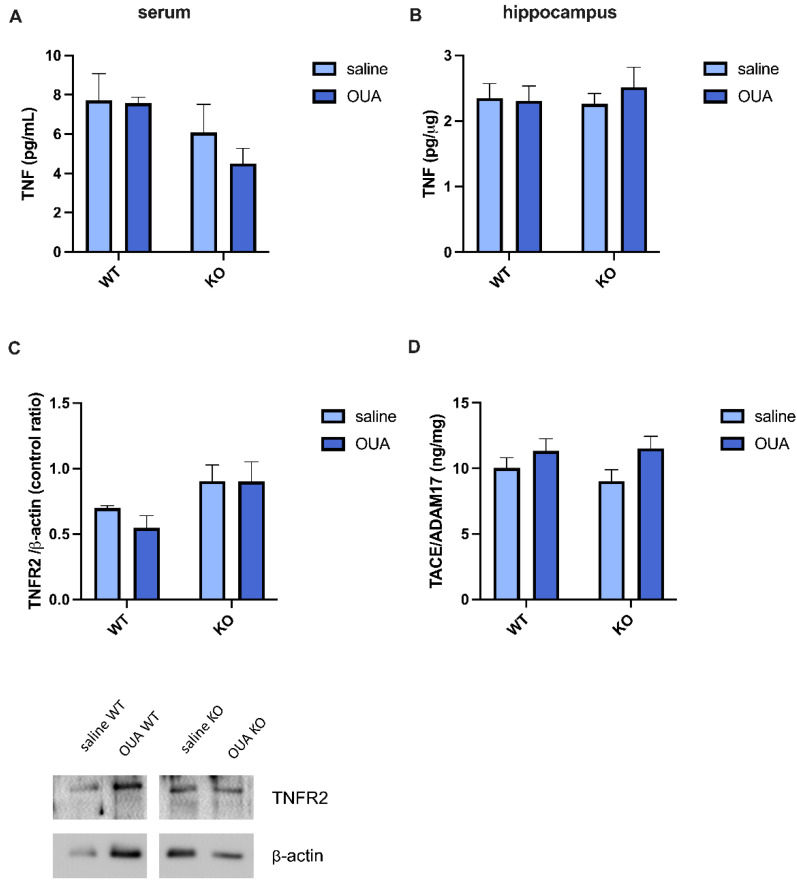
Effect of ouabain and the lack of TNFR1 in TNF levels, TNFR2 expression, and TACE/ADAM17 activity. (**A**) TNF levels in serum were not changed by ouabain treatment or the TNFR1 gene. (**B**) TNFR1 KO mice presented the same TNF levels as WT mice in hippocampus. (**C**) Densitometric analysis and representative Western blotting of TNFR2 membrane expression. There was an increase in TNFR2 expression in the membrane enriched fraction for the gene factor [F (1, 30) = 5.591]. (**D**) The activity of TACE/ADAM 17 was raised for ouabain treatment [F (1, 16) = 4563]. Results are expressed in pg/mL for serum samples (*n* = 11, N = 3) and pg/υg (mean  ±  SEM) in hippocampal samples (*n* = 4, N = 2) for TNF measurements, in control ratio for TNFR2 expression (*n* = 5, N = 2), and ng/mg in TACE/ADAM 17 activity (*n* = 5, N = 1). Two-way ANOVA was performed for the comparison followed by Tukey´s post-test.

**Figure 2 biomedicines-10-02937-f002:**
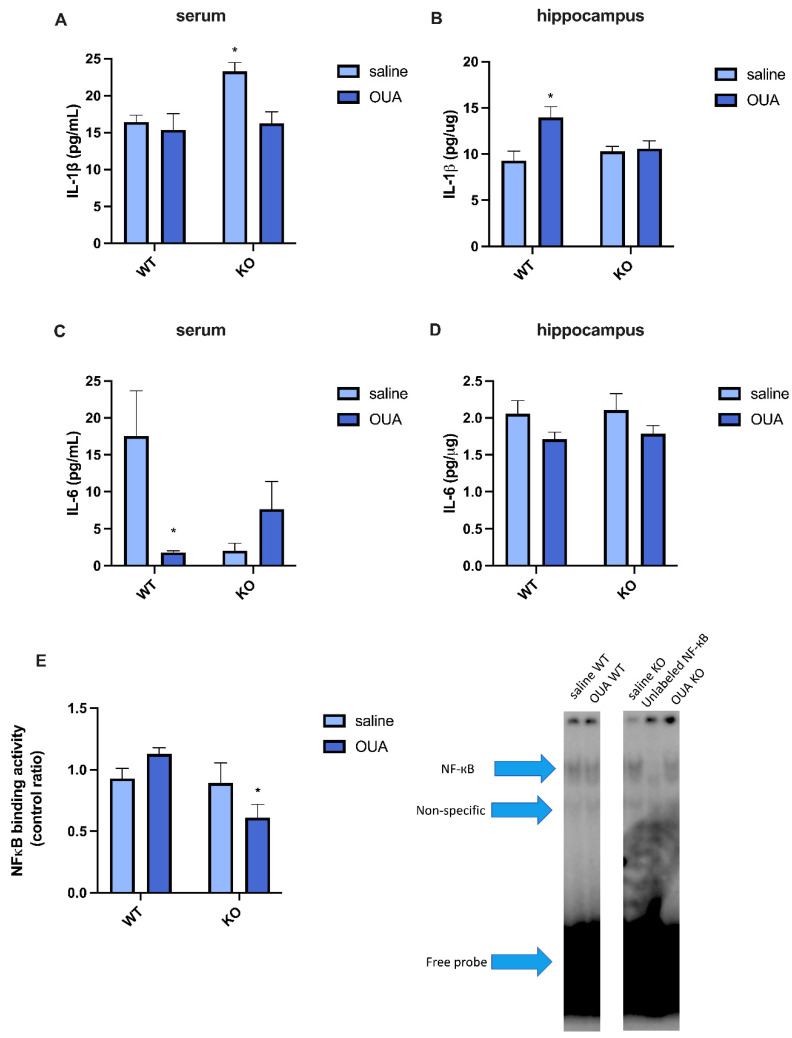
TNFR1 KO mice respond differently to ouabain in comparison to WT mice. (**A**) Basal levels of IL-1β in serum are higher in TNFR1 KO in contrast with the other groups, * vs. saline WT, OUA WT, and OUA KO (* *p* < 0.05). (**B**) Ouabain increases hippocampal IL-1β levels in WT mice, * vs. saline WT (* *p* < 0.05). (**C**) Ouabain reduces the basal serum levels of IL-6 only in WT mice, * vs. saline WT (* *p* < 0.05). (**D**) No changes were found in the IL-6 response in the hippocampus. (**E**) Densitometric analysis and representative EMSA of NF-кB binding activity showed that ouabain causes different responses in NF-кB activity according to the presence or not of TNFR1. Ouabain decreases the NF-кB activity in TNFR1 KO mice, * vs. OUA WT (* *p* < 0.05). Results are expressed in pg/mL for serum samples, pg/υg (mean  ±   SEM), or NF-кB binding activity (control ratio) in the hippocampus (*n* = 4, N = 2, only IL-1β in the hippocampus has three individual experiments). Two-way ANOVA followed by Tukey’s post-test.

**Figure 3 biomedicines-10-02937-f003:**
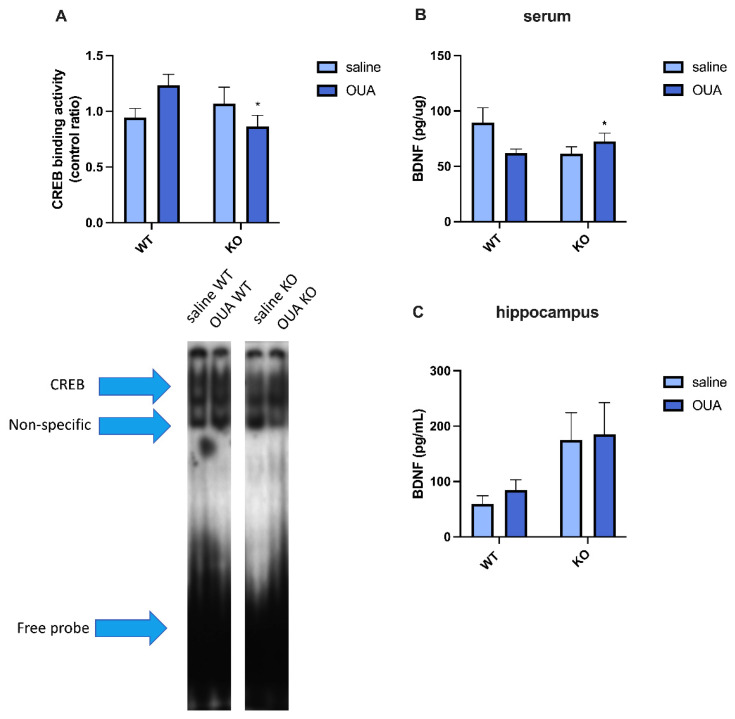
CREB activity and BDNF levels in the hippocampus and serum. (**A**) Densitometric analysis and representative EMSA of CREB binding activity showed an interaction between gene and treatment [F (1, 15) = 4987, (* *p* < 0.05)], the same interaction was seen in (**B**) for BDNF levels in the hippocampus [F (1, 16) = 4541, (* *p* < 0.05)]. (**C**) Levels of BNDF in serum are higher for gene factor [F (1, 18) = 7511]. Results are expressed in a ratio form control for CREB activity (*n* = 4, N = 2) and pg/υg (mean  ±  SEM) for hippocampal BDNF levels (*n* = 4, N = 2) and or pg/mL (mean  ±  SEM) for BDNF serum levels (*n* = 5, N = 2). The samples were analyzed by two-way ANOVA, followed by Tukey´s post-test.

**Figure 4 biomedicines-10-02937-f004:**
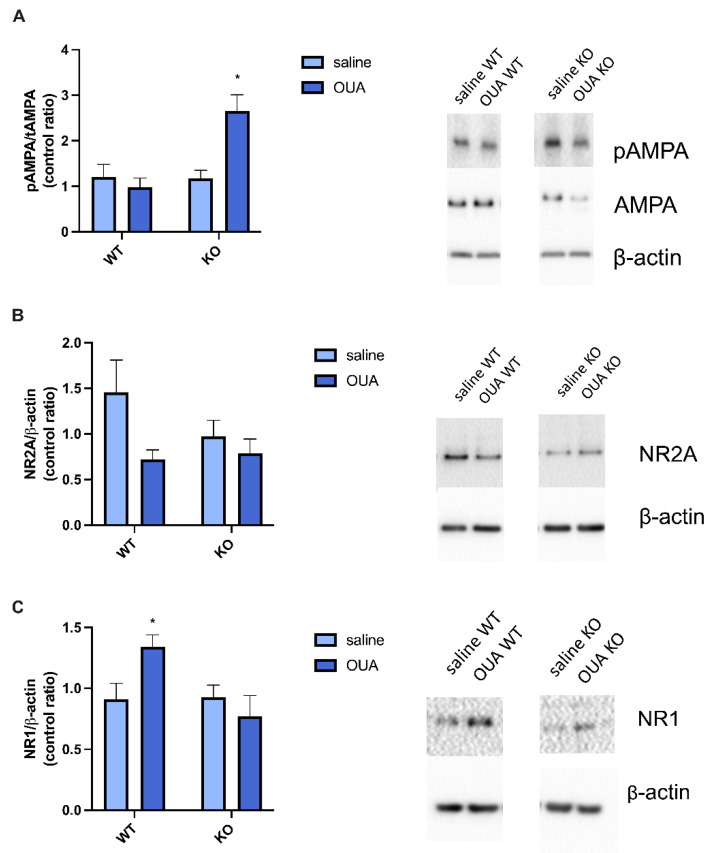
Ouabain causes changes in AMPA receptor phosphorylation, NR1 and NR2 membrane expression. (**A**) Densitometric analysis and representative Western blotting of phosphorylation ratio of AMPA receptor. Ouabain in TNFR1 KO mice is increased in comparison to all the other groups * vs. saline WT, saline KO, and OUA WT (* *p* < 0.05). (**B**) Densitometric analysis and representative Western blotting of NR2 membrane expression. NR2 expression is decreased for the ouabain factor [F(1, 14) = 5067]. (**C**) Densitometric analysis and representative Western blotting of NR1 membrane expression. Ouabain in WT mice increases NR1 levels in comparison to TNFR1 treated with ouabain, * vs. OUA KO (* *p* < 0.05). Results are expressed in the phosphorylation ratio for AMPA receptor (*n* = 4, N = 2), in control ratio for NR2 (*n* = 4, N = 2), and for NR1 (*n* = 7, N = 2). The samples were analyzed by two-way ANOVA followed by Tukey’s post-test.

**Figure 5 biomedicines-10-02937-f005:**
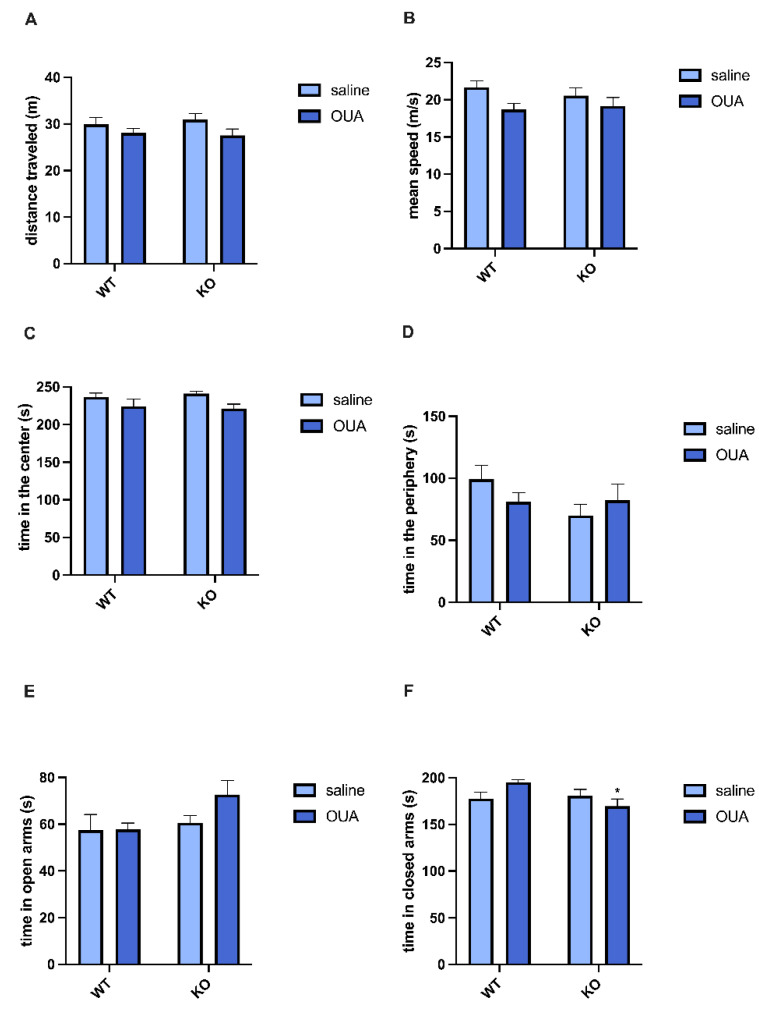
Ouabain causes changes in behavior even after 24 h (open field) and 48 h (plus-maze) after ouabain injection. (**A**) corresponds to the distance traveled, and there is a decrease in this parameter for the ouabain factor [F (1, 49) = 4175]. (**B**) represents the mean speed which is also decreased for the ouabain factor [F (1, 46) = 4967]. (**C**) corresponds to the time in the center that has a decrease for the ouabain factor [F (1, 48) = 5638]. (**D**) represents the time in the periphery that does not have any changes between the groups. (**E**) corresponds to the time in the open arms in the elevated plus-maze test, and no changes between the groups were found. (**F**) corresponds to the time in closed arms in the elevated plus-maze test, and ouabain decreases the time in closed arms in TNFR1 KO mice compared to WT treated with ouabain, * OUA WT (* *p* < 0.05). Results are expressed in seconds (mean  ±   SEM) for open field test (*n* = 10, N = 3) and for plus-maze (*n* = 9, N = 3). Only mean speed is expressed in meters per second (mean ±  SEM). The samples were analyzed by two-way ANOVA followed by Tukey’s post-test.

## Data Availability

Not applicable.

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
