# Peer review of "Consequences of the Lack of TNFR1 in Ouabain Response in the Hippocampus of C57BL/6J Mice"

_biomedicines, 2022, doi:10.3390/biomedicines10112937_

Round 1

Reviewer 1 Report (New Reviewer)

The authors describe the inflammatory response to the steroid hormone ouabain when TFNR1 is not missing, using a Knock-out mouse model. 

However, the statistical significance of the results represents a major concern. In particular, Figure 1 reports the statistics in the legend but not in the figure panels. Figure 2 reports the asterisks in the panels but it is not clear which comparison each asterisk refers to. Indeed, the Tukey's post hoc test implies that each condition is compared with all the others in a one to one fashion, therefore different p values are expected for each comparison. 

This applies also to all the following figures, whenever asterisks are shown in the graphs.

Figure 2 E right shows a representative EMSA with one extra unknown well between saline KO and OUA KO.

Figures 3B and 3C are exchanged in the figure legend. Additionally the interaction between gene and treatment mentioned in the legend is not evident from the graphs.

Figure 4 shows the same issue described above for the lack of statistics in panel B and the uneasy understanding of the asterisk in panels A and C.

Figure 5 describes the effect of ouabain on the mouse behaviour, but no significant effect is visible from the graphs.

Author Response

I would like to thank the reviewer for the comments that help us to improve the manuscript.

The authors describe the inflammatory response to the steroid hormone ouabain when TFNR1 is not missing, using a Knock-out mouse model. 

However, the statistical significance of the results represents a major concern. In particular, Figure 1 reports the statistics in the legend but not in the figure panels.

The issue is that the statistical difference is only in the two-way ANOVA. it is really hard to represent it in a way that the graph looks clean. Even the reviewers in the previous round said : “If this makes indicators of significance difficult, the main ANOVA effects can be reported in the figure legends.” and that is exactly what we did.

Figure 2 reports the asterisks in the panels but it is not clear which comparison each asterisk refers to. Indeed, the Tukey's post hoc test implies that each condition is compared with all the others in a one to one fashion, therefore different p values are expected for each comparison. 

In figure 2A : (A) Basal levels of IL-1β in serum are higher in TNFR1 KO in contrast with the other groups,*vs saline WT, OUA WT, and OUA KO (*p<0.05). – In this example, the p<0.05 is the same for all the groups. We don’t see the need to add all the values of the “ps” for each comparison when they are not statistically different. The F value was only used for the differences in two-way ANOVA.

This applies also to all the following figures, whenever asterisks are shown in the graphs.

Figure 2 E right shows a representative EMSA with one extra unknown well between saline KO and OUA KO.

The extra unknown sample is a unlabeled NF-kB sample that shows the specific band of NF-kB. The label for this sample was added in the figure.

Figures 3B and 3C are exchanged in the figure legend. Additionally the interaction between gene and treatment mentioned in the legend is not evident from the graphs.

The label in 3B and 3C was wrong. The figure was changed.

It is not evident in the graphs but there is a statistical difference according to GraphPad Prisma and as described by F values.

Figure 4 shows the same issue described above for the lack of statistics in panel B and the uneasy understanding of the asterisk in panels A and C.

The same as figure 2. As described in legend, according to previous reviewer solicitation, Panel A, Ouabain in TNFR1 KO mice is increased in comparison to all the other groups *vs saline WT, saline KO and OUA WT (*p<0.05). Panel C, Ouabain in WT mice increases NR1 levels in comparison to TNFR1 treated with ouabain,*vs OUA KO (*p<0.05).

Figure 5 describes the effect of ouabain on the mouse behaviour, but no significant effect is visible from the graphs.

It is not evident in the graphs but there is a statistical difference as F value indicates: F(1, 49) = 4,175, (*p<0.05) for item A,  F (1, 46) = 4,967,(*p<00.5)] for item B, and F (1, 48) = 5,638,(*p<00.5) for item C

Reviewer 2 Report (Previous Reviewer 1)

I am satisfied with the response from the authors and feel no additional changes are required. 

Author Response

I would like to thank for the reviewer comments.

Round 2

Reviewer 1 Report (New Reviewer)

Dear authors, 

thanks for clarifying about the statistical analysis.

However, the updated manuscript does not show yet the label in figure 2E reffering to the unnamed column, object of our discussion. Please include it.

Author Response

I am sending the new figure 2 with the label in figure 2E reffering to the unnamed column.

This manuscript is a resubmission of an earlier submission. The following is a list of the peer review reports and author responses from that submission.

Round 1

Reviewer 1 Report

In this manuscript, Kinoshita et al evaluate the impact of TNFR1 knockout on inflammatory signaling in the murine hippocampus at baseline and following peripheral administration of the cardiac glycoside ouabain, which the authors propose might be neuroprotective and anti-inflammatory at lower doses. The results presented are not groundbreaking but might be of potential interest and I am a proponent of the importance of negative data to the field. However, there are several issues with the data presentation, conclusions drawn, and writing of the manuscript that significantly lessen my enthusiasm for the report. Should the authors correct these issues, I would be open to seeing a heavily revised paper.

Major Concerns:

1.      The authors cite a paper in the introduction indicating that ouabain increase TNFa in the hippocampus of rats, but they do not observe the same effects in mice. Why?

2.      The labeling convention for graphs varies from figure to figure (e.g., Figure 1 you have saline vs OUA in grey/black bars and WT/KO on the x-axis labels vs the opposite in Figure 2 and 3). This is very confusing for the reader. Please use the same labeling scheme throughout the manuscript. If this makes indicators of significance difficult, the main ANOVA effects can be reported in the figure legends.

3.      The paragraph on lines 69-73 is confusing. I think I follow the logic, but it should be rewritten to state the following: 1) Chronic TNFa signaling can promote tissue damage. 2) Elevations in TNFa levels are found in the brains of patients with neurodegenerative disease (which ones????). 3) Treatment with neutralizing antibodies ameliorated symptoms (which ones???) in patients with neurodegenerative disease (which one). 4) They can’t be used in MS because of the risk for worsening demyelination. References for these statements should be provided.

4.      Please replace the paragraph on lines 93-96 with a central hypothesis for the work. This section is a little murky. Something like “Based on prior work, we hypothesized that ouabain might impact neuroinflammation via TNFa signaling and sought to test this hypothesis…”

5.      Please provide catalog information for the kits used to measure TNFa, IL-6, and IL-1Beta. There is some variability in sensitivity across reagents and this information is critical for replication.

6.      All biochemical analyses were performed 2 hours after injection, but the behavior was performed at 24 hours (open field) or 48 hours (EPM). What was the reasoning for these time points? What is the half-life of ouabain?

7.      Were the animals used for open field the same cohort used for EPM? If so, please provide a simple timeline indicating the number of mice in each group and the experiments performed.

8.      In the figure legends, the N for each experiment (all groups) should be provided. It is also unclear if the symbols of significance represent factor analyses or post-hoc comparisons. Please clarify. If major effects, provide the F values from the two-way ANOVA. For all graphs, you should indicate on the graph the groups being compared for post-hoc analyses and indicators of significance.

9.      The titles for each results section should mention the major conclusions from that section. Currently they are descriptive and provide no information about key results.

10.   The transition from section 3.1 to 3.2 is awkward. What is the justification for looking at IL-1b signaling? Any literature indicating that TNFR signaling controls IL-1b? Does ouabain impact IL-1b. References are needed here.

11.   The impact of ouabain vs TNFR1 KO on IL-1b in the periphery vs hippocampus is curious. This should be discussed. What is the hypothesized mechanism to explain these data?

12.   In Figure 2, Please label the graphs with the source of the data (hippocampus vs serum). Also, please switch 2C and 2D so that each set (IL-6 and IL-1b) are in the same order. This is confusing otherwise.

13.   In Figure 2D, OUA clearly decreases IL-6 in the WT mice, but it is unclear if 1) TNFR1 mice have lower levels of IL-6 or 2) what is the impact, if any, on IL-6 in the KO mice (it looks like no significant effect). Please clarify the description of these data.

14.   For Figures 3A and 3B, it is noted that there was a significant interaction effect but no indicators of significant changes. Please be more specific. If no post-hoc tests were significant, this should be noted.

15.   In figure 4A, the authors claim that AMPA phosphorylation is elevated, but this is misleading. Based on their representative blots they actually see a decrease in total AMPA expression (thus an increase in pAMPA/AMPA). Thus, it is more correct to state that AMPA expression is decreased leading to an increase the in proportion of phosphorylated receptors.

16.   The figure legend for Figure 5 is an overstatement. I don’t see “important changes in behavior”. The impacts of genotype or treatment are modest at best. As stated above, the authors need to justify why they used 2 different time points for behavior vs biochemical experiments. Are there prior papers indicating that the impacts of OUA on these behaviors persist for 24-48 hours?

17.   Why were open field and EPM chosen as behaviors? Given the focus on hippocampus and glutamatergic signaling it would be prudent to instead look at measures of hippocampal-dependent memory.

18.   For the EPM data, if the authors are only quantifying time in open vs closed arms these patterns should be mirrored. I assume that the authors quantified time in center separately. This should be noted in the materials and methods section.

19.   It would appear that many studies utilize icv injection of ouabin to modulate animal behavior. These authors, however, use an ip injection. What is the bioavailability of OUA in the brain after IP administration? Clearly it has some effect, but the pharmacokinetic profile of the drug will be vastly different for these two routes of administration. This should be included in the discussion as a limitation to the current study. It is possible, for example, that icv injection of ouabain might alter TNF signaling in mice.

20.   The discussion needs a complete overhaul. It is lacking in proper references, largely descriptive, and highly redundant. The following topics should be included:

a.      Specific examples of increased neuroinflammation in neurodegenerative diseases. For which diseases has an increase in TNFa been observed?

b.      The clinical use of TNFa neutralizing antibodies in neurodegeneration (what diseases? References?)

c.      The complexity of TNF signaling via TNFR1 vs R2. How do TNFR1 and TNF2 impact neuronal signaling (specific mechanisms!)? What effects are beneficial and what effects are deleterious in the context of neurodegeneration?

d.      What is the proposed mechanism of Ouabain in neuroprotection? What do the results from this paper tell us about ouabain’s impacts in the hippocampus and the role of TNFa?

e.      How does ouabain impact TNF signaling? Prior work and results here. Why the disconnect? Is there evidence of differences in immune responses between rats and mice? Are these effects through NKA inhibition??? If so, why was a dose chosen that will not impact this target? This is unclear.

f.       The major conclusions of the paper are listed here. Some are discussed, some not. Some have appropriate references, and some don’t. This should be more systematically described. Elements of the discussion not related to these key elements can be eliminated. The discussion should discuss the results, not repeat the impetus for each experiment (that’s the introduction) or simply re-state the results. The discussion could easily be cut by half.

                                                    i.     TNFR1 KO leads to an increase in TNFR2 in brain (though not clear if this is hippocampus). Why? How might this impact neurons?

                                                   ii.     TNFR1 KO renders mice insensitive to OUA-dependent changes in IL-1b and IL-6. What is the mechanism? How might this impact neurons?

                                                  iii.     Ouabain increases TACE. How might this impact neurons?

                                                  iv.     TNFR1 KO renders mice insensitive to the impacts of ouabain on NMDA receptor subunits. Why might this be important? What is the mechanistic connection between TNFR and expression of these proteins?

                                                   v.     There are a few opposing actions of TNFR1 in the hippocampus vs periphery

1.      TNFR1 KO increases IL-1b and decreases IL-6 in the periphery but not hippocampus. Why might this be? Are there differences in IL-1b production/release between the peripheral immune system and microglia? Other sources in the brain? How might this impact neurons?

2.      TNFR1 KO decreases BDNF in the hippocampus but increases it in serum. Why? Mechanism? Functional importance?

Grammatical errors:

1.      Line 35, delete “described as a”

2.      Line 37, “high” should be “highly”

3.      Line 38, Delete both commas. Insert “that” before “plays”

4.      Line 39, “ouabain” should be “ouabain’s”

5.      Line 52, delete “besides”

6.      Line 76, replace “preferably for” with “with a preference for”

7.      Rewrite the sentence on lines 78-79 to be “TNFR1 and TNFR2 may have antagonistic or synergistic actions depending on the context”

8.      Line 83, replace “that results in different signaling pathways activation” with “and, as a result, activates different pathways”

9.      Line 87, delete “besides”

10.   Line 126, there should be no new sentence between “ph 7.4) and “resulting”

11.   Line 206, replace “which its” with “whose”

12.   Line 296, “mediator” should be “mediators”

Reviewer 2 Report

Comments and suggestions

1. Title should be written more accurately according to your study that is missing, such male mice? C57BL/6J missing

2. In case of abbreviation first-time short form with abbreviation then use short form throughout the manuscript.

3. Abstract should be more concise.

4. In the introduction there is no clear objectives of your study. You should write the specific aim of your study at the end of the introduction.

5.  In the introduction, Ouabain is a cardiotonic steroid described as a hormone produced in the adrenal and 35 pituitary gland (Hamlyn and Blaustein, 2016; Kawamura et al., 1999; Murrell et al., 2005; 36 Nesher et al., 2007; Schoner, 2000). Ouabain interacts with Na+ , K+ - ATPase (NKA), a high 37 conserved membrane protein, plays an important role in the cell osmotic balance, and is 38 critical for neuronal excitability (7,8); Please check the wrong formatting.

6.  Materials and Methods section

2.1. Animals and treatment should be revised correctly.

7. Chemicals and ELISA kits; Not written correctly. 

8.  Figure 2, check it has been compared between two groups. How many mice each group?

9. Figure 3 figure not clear. Please provide a better resolution.

General comments: Formatting and a lot of typos, and grammatical errors throughout the manuscript that should be revised with the help of experts.